



# Monitoring tidal hydrology in coastal wetlands with the 'Mini Buoy': applications for mangrove restoration

Thorsten Balke[1], Alejandra Vovides[1], Christian Schwarz[2], Gail L. Chmura[3], Cai Ladd[1], Mohammad Basyuni[4]

[1] School of Geographical and Earth Sciences, University of Glasgow, Glasgow, G128QQ, United Kingdom
[2] School of Marine Science and Policy, University of Delaware, Lewes, 19958 DE, United States
[3] Department of Geography, Mc Gill University, Montreal, QX H3A 0B9, Canada
[4] Department of Forestry, Faculty of Forestry, Universitas Sumatera Utara, Medan 20155, Indonesia

*Correspondence to*: Thorsten Balke (thorsten.balke@glasgow.ac.uk)

## Abstract

Acquiring in-situ data of tidal flooding is key for the successful restoration planning of intertidal wetlands such as salt marshes and mangroves. However, monitoring spatially explicit inundation time series and tidal currents can be costly and technically challenging. With the increasing availability of low-cost sensors and data loggers, customized solutions can now be designed to monitor intertidal hydrodynamics with direct applications for restoration and management.

In this study, we present the design, calibration, and application of the Mini Buoy, a low-cost bottom-mounted float containing an acceleration data logger for monitoring tidal inundation characteristics and current velocities derived from single-axis equilibrium acceleration (i.e. logger tilt).

The acceleration output of the Mini Buoys was calibrated against water-level and current velocity data in the hypertidal Bay of Fundy, Canada, and in a tidally reconnected former aquaculture pond complex in North Sumatra, Indonesia. Key parameters, such as submersion time and current velocities during submergence can be determined over several months using the Mini Buoy. An open-source application was developed to generate ecologically meaningful hydrological information from the Mini Buoy data for mangrove restoration planning. We present this specific SE Asian mangrove restoration application alongside a flexible concept design for the Mini Buoy to be customized for research and management of intertidal wetlands worldwide.



# 1 Introduction

## 1.1 Tidal flooding


Tidal flooding is the main driver of species distribution and ecosystem functioning in intertidal wetlands (i.e. salt marshes and mangroves). Inundation determines the composition of plant communities through physiological inundation and salt-tolerance limits, soil anoxia and altered competitive ability along the tidal flooding gradient (Bockelmann et al., 2002; Jiménez et al., 1991; van Loon et al., 2007; Matthijs et al., 1999; Pennings and Callaway, 1992; Silvestri et al., 2005). Tidal currents can

disperse propagules (Delgado et al., 2001; Koch et al., 2009; Van der Stocken et al., 2013; Wiehe, 1935) and dislodge seedlings when drag forces exceed the stability threshold of rooted seedlings (Balke et al., 2011, 2014; Wiehe, 1935; Xiao et al., 2016). Interactions between tidal currents and the friction created by vegetation (Bouma et al., 2013; Schwarz et al., 2018; Temmerman et al., 2007) determine the evolution of intertidal landscapes through flow routing and altered sedimentation patterns (i.e. biogeomorphic feedbacks). Monitoring flooding gradients and tidal currents is not only key to understanding

intertidal wetland functioning but is also a prerequisite for the design and monitoring of successful restoration projects in the intertidal zone (Lewis, 2005; Primavera and Esteban, 2008; Wolters et al., 2005). Mangrove restoration projects in particular have shown low success rates due to the lack of hydrological site assessments prior to restoration (Primavera and Esteban, 2008; Wodehouse and Rayment, 2019). Assessment of the local hydrology prior to planting is needed to determine whether conditions are too harsh for seedlings to survive and need to be mitigated (Albers and Schmitt, 2015) or whether insufficient

flooding may lead to hypersalinity or succession towards terrestrial plant communities (Lewis, 2005). However, tidal current velocity monitoring in coastal wetlands is often limited in spatial coverage and duration or missing completely due to the high costs of monitoring equipment. Here, we present the design, calibration and application of a low-cost bottom mounted float equipped with an acceleration data logger (from here on referred to as the Mini Buoy) to monitor inundation duration and current velocities in intertidal environments with a specific application for SE Asian mangrove restoration.

## 1.2 Monitoring tidal inundation and currents

Key hydrological parameters in tidal wetlands are tidal inundation duration and inundation frequency (e.g. to select suitable species and sites for planting according to the flooding regime) and the duration of individual inundation free periods (also called Windows of Opportunity (WoO)) (Balke et al., 2011, 2014). WoO exceeding species-specific minimum durations allow for seedling establishment where disturbance (i.e. uprooting and erosion by waves and currents) is a limiting factor and hence

are important to assess in unassisted restoration projects (i.e. for natural colonization by pioneer species). Tidal current velocities are generally characterized by their flood and ebb tide magnitude to assess tidal exchange in enclosed systems and the potential for sediment transport or resuspension. Monitoring hydrodynamics in intertidal environments needs to be spatially explicit (i.e. across an elevational gradient or across a potential restoration site) to assess habitat suitability of a restoration site.





Inundation frequency and duration of tidal wetlands are generally derived from nearby tide-gauges or in-situ pressure-sensor deployments. The data is then projected onto digital elevation models or transects derived from levelling. Tidal current velocity is typically monitored using Electro Magnetic Flow Meters (EMF), Acoustic Doppler Velocimeters (ADV), or Acoustic Doppler Current Profilers (ADCP) (see for example Mullarney et al., 2017). The need for existing monitoring infrastructure and specialist equipment and knowledge means that capacity for hydrological and hydrodynamic monitoring may not always be readily available for restoration practitioners. Hydrodynamic time series data are also often limited in their duration due to the high demand for storage and battery capacity or due to concerns over theft or damage during storms. Hence long-term variability, rare disturbance events or stochastic WoO in coastal wetlands are often missed.

Proisy et al., (2018) note that large-scale mangrove restoration across disused and abandoned aquaculture ponds requires the assessment of elevations and hydrological conditions in each individual pond, since conditions can differ greatly depending on the site history. Hydrological site assessments in such complex anthropogenic tidal landscapes can thus quickly become costly and time demanding. Mangrove restoration manuals do acknowledge the need to assess site-specific tidal flooding and hydrodynamic exposure and propose practical 'low-tech' solutions for their quantification. A manual published by the Mangrove Action Project, for example, advocates measuring the elevation of a restoration site (using optical levelling equipment or a hose water level) against known references such as local tide gauges or elevations of nearby mangrove stands (Lewis and Brown, 2014). Semi-quantitative methods to assess hydrodynamic exposure, such as gypsum clod cards, are also proposed (Lewis and Brown, 2014). These methods are low-cost and locally implementable but not all are quantitative, scalable or comparable across mangrove environments. We propose that monitoring tidal inundation and hydrodynamic exposure using accelerometer technology could be an affordable, globally available and quantitative method to assess hydrological site suitability for mangrove restoration.

### 1.3 Accelerometer technology to monitor tidal currents

Accelerometer technology is widely used, from measuring shock and triggering safety air bags in the transport industry (MacDonald, 1990) to quantifying the physical activity of humans (Beanland et al., 2014; Passfield et al., 2017) and analysis of particle movement in geosciences (Akeila et al., 2010). Equilibrium acceleration data from two axes relative to the gravity vector can be used for precise tilt measurements of objects. This principle is used in dip-current meters which relate the tilt of a bottom-mounted float with current velocities and the drag forces exerted on the float (Figurski et al., 2011; Hansen et al., 2017). A similar single axis acceleration tilt calculation principle (i.e. the acceleration relative to the horizontal plane relates to the sine of the tilt angle) was applied here to determine inundation of the Mini Buoy and to correlate equilibrium acceleration of the Mini Buoy with current velocities (Fig. 1). For a detailed description of the individual forces acting on bottom-mounted floats see Figurski et al., (2011). Commercial dip current meters have recently become available however, to our knowledge, do not routinely include the analysis of flooding characteristics for intertidal applications and are still costly compared to the described setup with off-the-shelf components. Moreover, commercial dip current meters with a fixed calibration cannot be





customized. Adding additional sensors (as shown here for example a temperature sensor) or changing the shape and size of the float and length of the tether offers flexibility for the application in different environments. The single axis tilt calculation principle offers the benefit of reduced data storage requirements (i.e. increased deployment durations) and the potential

application of more economic sensors.

Here we describe the Mini Buoy, primarily as a design concept for the low-cost monitoring of abiotic conditions in periodically inundated intertidal ecosystems. We further present a fully calibrated Mini Buoy design with an open-source online data analysis application to assess tidal flooding (inundation and current velocities) for ecological mangrove restoration planning in SE Asia.


## 2 Methods

### 2.1 The Mini Buoy setup

The Mini Buoy consists of three readily available 'off the shelf' components: an acceleration data logger, a self-standing 50

ml centrifuge tube (CentriStar, Corning) and three fishing swivels. (See Fig 1a and A1 for a detailed description, overall weight = 42.3 g, displaced volume = 70 ml, and length of tube + swivels = 21.5 cm). Additional loggers can fit on top of the tube such as the HOBO MX2202 temperature and light sensor (see photo in Fig. 1 as used in the Bay of Fundy deployment). The Mini Buoy is anchored near the sediment surface and will float up when submerged and dip into the current according to the velocity magnitude near the bed. The acceleration data logger inside the Mini Buoy records gravitational acceleration as G-forces (g)

along the 3 axes of the tube relative to the gravity vector. Only y-axis acceleration was used in this study, hence any single-axis acceleration logger can be used for the Mini-Buoy. The MSR 145 B4A accelerometer (MSR Electronics GmbH, Switzerland) used in this study has a capacity of 2M measurements and an internal rechargeable battery. The data recording for x-axis and z-axis acceleration can be inactivated to increase storage capacity of the device. The sensitivity of the logger was set to the 2 g. The USB connector of the logger was facing upward in the tube, hence y-acceleration was 0g when the

logger was stationary in a horizontal (not submerged) position, and -1g when (submerged) in vertical position (Fig. 1). The time-averaged (i.e. equilibrium) y-axis acceleration output of the sensor ($Yacc$) corresponds to the tilt angle ($\theta$) of the sensor relative to the horizontal plane (i.e. orthogonal to the gravity vector) following Eq. (1):

$$Yacc \ [\text{g}] = -1\text{g} \times \sin\left(\theta\right) .$$

Y-axis acceleration was acquired every second for short-term deployments (several days) and every 10 seconds for long-term

deployments (single deployments of approx. 5 months are possible). We have chosen the described setup as the loggers, tube,




and tether are globally available, and the selected data logger itself has a waterproof housing and sufficiently large internal data storage for deployments across multiple spring neap tidal cycles.

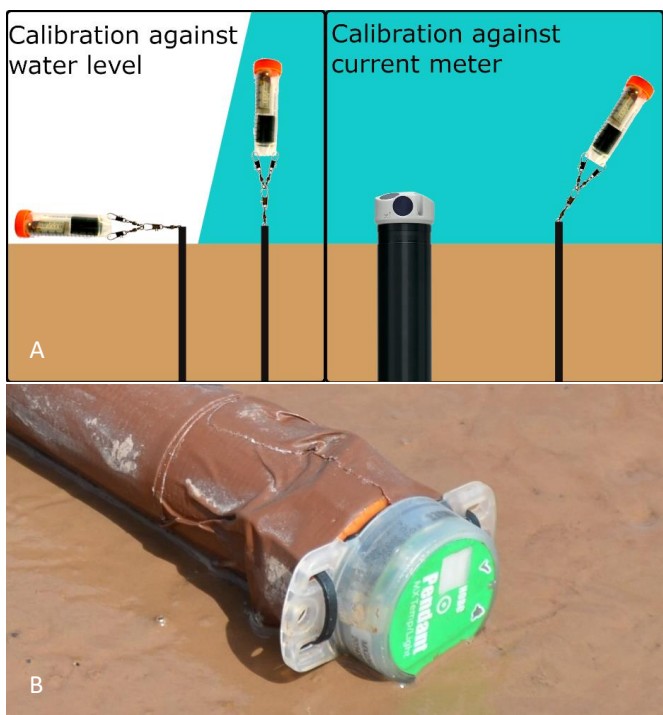


**Figure 1: A)** Movements of a small bottom mounted float (the Mini Buoy) equipped with an acceleration data logger during high and low tide. Single-axis equilibrium acceleration (i.e. tilt) of the buoy was calibrated against water-level and current-velocity data using an Acoustic Doppler Current Meter (pictured). The Mini Buoy consists of a flat-bottom centrifuge tube which is tethered to the tidal flat using fishing swivels and a metal stake. The acceleration data logger is a self-contained unit with internal battery and storage. Single-axis equilibrium
acceleration (with y axis acceleration =-1g when vertical i.e. flooded 0g when horizontal i.e. not flooded) correlates with the sine of the datalogger tilt relative to the horizontal plane **B)** Additional data loggers for temperature and light can be attached on top of the Mini Buoy. Further details on dimensions and logger specifications are available in the appendix.

## 2.2 Field sites

### 2.2.1 Bay of Fundy (Canada)

A short-term deployment of 9 days using 8 Mini Buoys (L1-L8) measuring acceleration at 1 Hz was carried out along a tidal-flat to salt marsh transect in the Bay of Fundy, Canada (45°47'6.80"N, 64°37'17.30"W). The Mini Buoys in the Bay of Fundy were also fitted with a HOBO MX2202 temperature and light sensor on top of the centrifuge tube (Fig. 1b). Measurements commenced on the 9th of June and ended on the 17th of June 2018. A Nortek Aquadopp 300 m current meter was installed 5 m seaward of Mini Buoy L1 for calibration. In addition, a pressure sensor (In-Situ Rugged Troll) measured continuously at 1
Hz and on-site barometric pressure correction was carried out using a second logger. The elevations along the transect were surveyed using an Emlid Reach View Differential GPS Rover with Real-Time Kinematic positioning. The vegetation along





the transect was <10 cm in height and sparse enough for the logger to rest horizontally on the sediment surface when not inundated (L4-L8 are located within the marsh). Metal rods were used to anchor the Mini Buoys in place. This site serves as a test of the Mini Buoy design concept in very energetic hypertidal conditions.


### 2.2.2 North Sumatra (Indonesia)

A short-term deployment of 7 days was also carried out along the East coast of North Sumatra (Fig. 2b, Percut Sei Tuan, Deli Serdang) starting on the 27[th] of November 2019. 15 Mini Buoys (B2-B16) measured acceleration at 1 Hz inside an abandoned, naturally recolonizing, aquaculture pond (3°43'22.86"N, 98°46'15.29"E). This pond is connected through constructed tidal

channels to the main estuary and partially recolonized by *Avicennia* spp., *Rhizophora* spp. and *Nypa fruticans*. This site was transformed into aquaculture ponds in 2002 and subsequently abandoned. One main channel directly connects the study pond to the surrounding ponds and the estuary through a previously breached embankment. The channel was equipped with an additional Mini Buoy (B1) which was calibrated using an upward-facing Nortek Aquadopp 300 m current meter located 5 m North of Mini Buoy B1 (Fig. 2b, also see 'channel' deployment in Fig. 6). The logger elevations of B9, B11, B12 and B14 in

relation to the pressure sensor of the current meter were established using a Leica optical level. An aerial image was acquired using a DJI Phantom 4 and Mini Buoy locations were mapped using ground targets. An additional short-term Mini Buoy was placed on a tidal flat facing the open coast directly seaward of colonizing *Avicennia marina* seedlings ('Open coast' deployment in Fig. 6 (3°44'1.55"N, 98°46'31.34"E).

Finally, long-term deployments of approximately 50 days were conducted at three locations: i) North Sumatra at an abandoned

pond in Percut Sei Tuan, where tidal regulation with a sluice system was still partially active (deployment 'Abandoned 1' in Fig. 6: 3°43'33.77"N, 98°46'22.70"E, 07.10.2019), ii) at the Percut Sei Tuan short-term study site described above (see 'long-term' logger nested within the short term deployment in Fig. 2b) which was exposed to regular tidal inundation ('Abandoned 2' in Fig. 6: 07.10.2019-26.11.2019), and iii) at a recent restoration site at Belawan with active planting of *Rhizophora apiculata* in May 2019 ('Restoration' in Fig. 6: 3°45'15.74"N, 98°42'22.43"E, 09.10.2019- 26.11.2019).








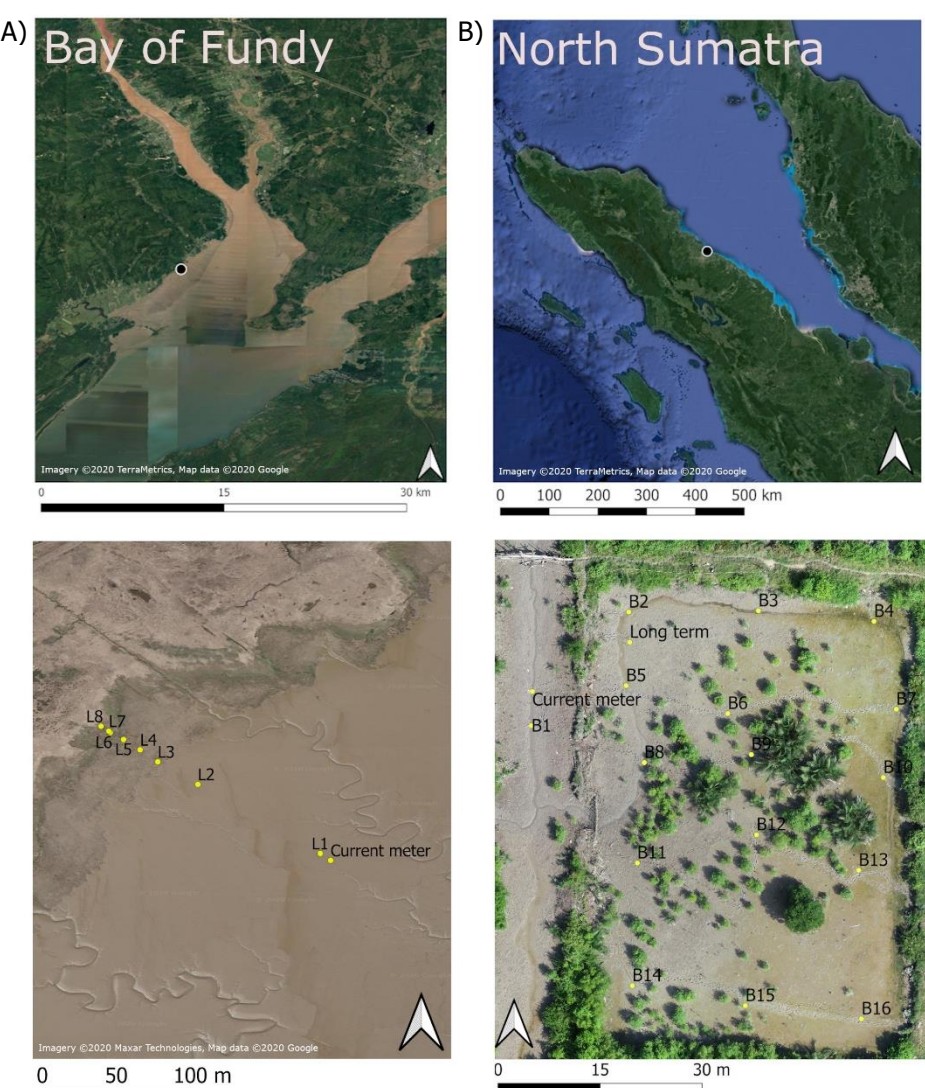

**Figure 2: A) The study sites in the Bay of Fundy, Canada with 8 Mini Buoys + temp/light sensor (see Fig. 1b) deployed along an**
**elevational transect. L1 and L2 are located on the tidal flat, L3 at the marsh edge, and L4-L8 are within the salt marsh (bottom left).**
**B) The short-term study site in Percut Sei Tuan, North Sumatra, Indonesia, with 16 Mini Buoys (B1-B16) deployed across an**
**abandoned aquaculture pond that was naturally re-colonized by mangrove plants (drone image at bottom right). Both study sites**
**were equipped with a Nortek Aquadopp current meter close to L1 and B1 to calibrate the Mini Buoys. Maps were created using**
**QGIS.**





### 2.3 Training dataset to predict Mini Buoy inundation


Linear Discriminant Analysis (LDA) with R packages MASS (Venables and Ripley, 2002) and klaR (Weihs et al., 2005) was used to differentiate inundated time steps from non-inundated time periods using the acceleration data. The LDA is a machine learning tool that requires a training dataset (i.e. inundation time series at the Mini Buoy location) to then predict inundation events for other Mini Buoys and time periods. This approach was chosen as it can be directly applied to other Mini-Buoy

designs that may have different buoyancy, dimensions and data loggers. Mini-Buoys closest to the Aquadopp current meter and pressure sensor (L2 - L5 at the Bay of Fundy and B9, B11, B12 and B14 in North Sumatra) were selected to generate a training dataset at 1-minute resolution containing flooded and non-flooded classes based on water-level and elevation data. For North Sumatra, water-level data was linearly interpolated to 1-minute frequency using the Aquadopp pressure sensor diagnostic output, and water-level data was generated directly from a separate pressure data logger at 1-minute frequency at

the Bay of Fundy. The Mini Buoys were considered 'flooded' when fully submerged (i.e. bed elevation + 20 cm). The LDA then separated and predicted flooded and non-flooded cases using 1-minute median y acceleration and the difference between the 1-minute 75[th] and 25[th] percentile of the y acceleration (i.e. the variability of y axis measurements) as predictor variables. The klaR package provides apparent error rates (i.e. the proportion of observed cases incorrectly predicted), and additional cross validation using LDA predictions for a subset of Mini Buoys was carried out. The generated LDA's were used to predict

all Mini Buoy inundation classifications (L1-L8 and B2-B16) to create time series of inundation characteristics and subsequently predict current velocities.

### 2.4. Mini-Buoy calibration against current velocity

At both study sites an Aquadopp 300 m current meter (Nortek) was installed with the measurement head 20 cm above the

sediment surface. The blanking distance was set to 35 cm and the device measured in diagnostic mode with 1025 sample bursts every hour at 1 Hz (internal sampling rate = 23 Hz). Median y acceleration over a 1-minute period for both Mini Buoys (L1 at Bay of Fundy and B1 in North Sumatra) was correlated with the median current velocities for both horizontal velocity axes over a 1-minute period.

The site and design-specific calibration curves were further used to predict current velocities for all deployed Mini Buoys (Fig. 2). In order to separate individual high-tide events for each Mini Buoy, inundation events were separated whenever no inundation occurred for at least 100 minutes. As partially inundated Mini Buoys during early phase of submergence or late stages of emergence may create inaccurate velocity predictions, a buffer time period at the start and end of each high tide was implemented. The first and last 10 minutes of each predicted tide for Bay of Fundy data and 50 minutes for the North Sumatra

time series were not used for velocity predictions. This buffer duration was based on the time it takes for the tide to rise and fall by 20 cm (i.e. the Mini Buoy height) at each site, as estimated from water level measurements. Inundation durations were





calculated from the predicted inundation categories using the LDA, and each high tide event was separated into two equal halves to separate approximate flood and ebb tides. Overall current velocities for the entire monitoring period were predicted as median values and 75 percentiles to reduce the influence of outliers (e.g. due to passing boats, waves or turbulence).

**2.4 Long-term deployments and mangrove restoration site assessment application**

Acceleration data acquisition can be reduced to 10-second intervals to allow for longer deployment periods across spring-neap tidal cycles and seasons. For this purpose, Mini Buoys B1, B11 and B12 were re-sampled at 10 second intervals to construct a new low-frequency calibration. The same analyses for inundation predictions using the LDA and correlation against current velocities (described above) were carried out over 15-minute time periods. This analysis was only performed for the North

Sumatra case study to develop a monitoring application for mangrove restoration in SE Asia. Mini Buoys were considered submerged in the calibration time series when measured water levels were above the Mini Buoy for the entire 15-minute period.

An online app, using the R package shiny (Chang et al., 2020), was created as a quick and easy way to assess local hydrological

site conditions prior to restoration of mangroves. Instead of the LDA, a fixed acceleration threshold was applied to differentiate between flooded and non-flooded time steps. This threshold allows for the use of the Mini Buoy and the R shiny application without the need to load training datasets and was informed by the results of the LDA. The fixed threshold predictions were further cross validated against the LDA predictions for Mini Buoys B11 and B12. The application calculates inundation and current velocity statistics for the entire monitoring period: average high tide duration (min), average flooding duration (min/d),

flooding frequency ($d^{-1}$), maximum WoO duration (d) = longest inundation free period, median current velocity (m/s), 75 percentile current velocity (m/s), and difference of flood - ebb median velocity (m/s). The R shiny application separates individual inundation events (i.e. high tides) using a minimum inundation free period of 100 minutes and removes the first and last 60 minutes of each inundation event for current velocity predictions to avoid misinterpretation of velocities in very shallow or stagnant water. The application also allows for a comparison with a second Mini Buoy to assess differences in conditions

between a restoration target site and a reference site which has the desired conditions/habitat (i.e. previously successful restoration site or site with natural colonization of the desired species). The R shiny app code and a Mini Buoy handbook for logger programming, field deployment and data analysis are available via Github. The app can also be used directly online via https://mangroverestoration.shinyapps.io/MiniBuoyApp/ and the handbook can be downloaded from zenodo: https://doi.org/10.5281/zenodo.4245198.




## 3 Results

### 3.1 Flooding and current velocity calibration

The LDA separated the Mini Buoy data in inundated and non-inundated time steps with apparent error rates of 0.008 for merged L2-L5 Mini Buoy data (Bay of Fundy) and of 0.015 for merged B9, B11, B12 and B14 Mini Buoy data (North Sumatra) (Fig. 3a). Using Mini Buoy data of L2 and L3 as a training dataset to predict L4 inundation classification with the LDA at 1-minute intervals at the Bay of Fundy transect produced 1.57% false predictions. Cross validating predictions with an LDA generated from B11 and B12 to predict B14 at the North Sumatra site produced 1.4% false predictions when validated against water level classification. The variability of the y-axis acceleration within the 1-minute time step (i.e. differences between the 75th and 25th percentile of y acceleration) was different between both field sites. Percentile ranges of up to 0.5 g were recorded at the Bay of Fundy, the Mini Buoy within the tidal channel in North Sumatra showed values of up to 0.08 g. This is likely attributed to the more dynamic conditions at the Bay of Fundy where significant wave heights calculated from the Aquadopp data reached 0.73 m during the monitoring period. Mini Buoys deployed further away from the pressure sensor were not considered for calibration as the local water levels may be influenced by local flooding and draining pattern through tidal channels.

Median current velocities for 1-minute time steps at both locations correlated with median y-axis acceleration over the same time step and were best described using a 3rd order polynomial regression rather than a sine function (Fig. 3b). The polynomial fit may be attributed to a change in the proportion of the sampled velocity profile (i.e. the Mini Buoy dips into faster currents and hence stays closer to the bed when compared to low currents). The calibration curves for both field sites show that current velocities above ~0.1 m/s could be measured by the Mini Buoy. The Bay of Fundy Mini Buoys fitted with a temperature and light logger showed a slightly higher sensitivity to low velocities (likely due to reduced buoyancy). Velocities measured at the Bay of Fundy reached up to 0.5 m/s whereas velocities at the North Sumatra tidal channel stayed generally below 0.4 m/s. After applying the identification of individual tides and removing the buffer time periods at the start and end of each inundation event, continuous predictions of velocities were achieved (see Fig. 4 for examples of tidal current predictions). Averaged tidal current predictions and inundation durations were plotted across the elevation transect at the Bay of Fundy and showed a rapid decrease in median current velocities with elevation (Fig. 5a). An increase in median current velocities is observed at the edge of the highest salt marsh terrace compared to the lower marsh elevations. Animations of the predicted inundation events and current velocities are available from the online materials.





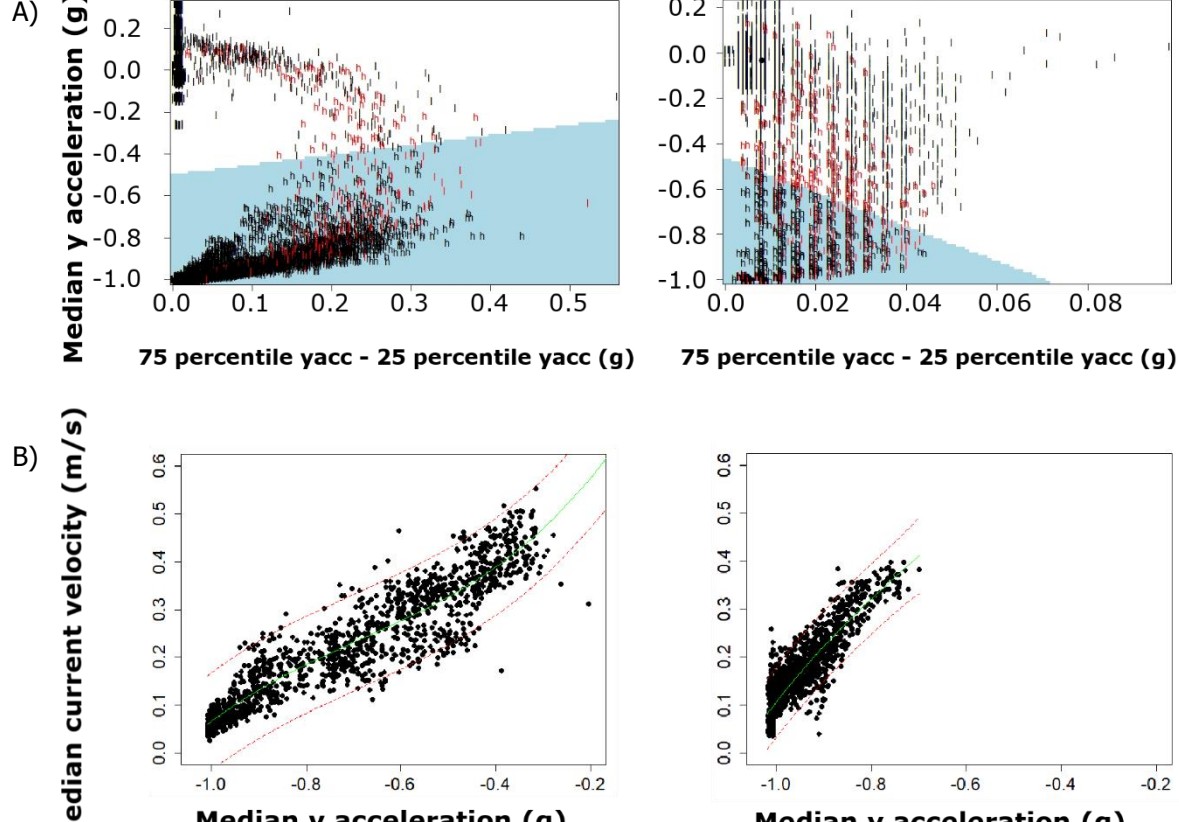

Figure 3: Training data for the Mini Buoy calibration at both study sites. A) Linear Discriminant Analysis (LDA) was used to determine the time-averaged acceleration signal of the y-axis for flooded and non-flooded time steps of L2-L5 (Bay of Fundy, left) and B9, B11, B12, B14 (North Sumatra, right). The error rate of the LDA for the Bay of Fundy was 0.008 and for North Sumatra 0.015 (l=low tide, h=high tide, red letters = false predictions). The blue-shaded area indicates the range of values for which inundation is predicted by the LDA. The LDA was then used to predict inundation events for all Mini Boys.

B) Polynomial regression provided the best fit for the median y-axis acceleration [$Y_{acc}$ (g)] against the measured current velocity [$V_{cur}$ (m/s)] for 1-minute intervals at both study sites. L1 Bay of Fundy: $V_{cur} = Y_{acc}$ 0.8812+(1.9209* $Y_{acc}$)+((2.1394* $Y_{acc}$ ^2)+((1.0344)* $Y_{acc}$ ^3) , $R^2$adj.=0.84, P<0.05; B1 North Sumatra: function $V_{cur} = Y_{acc}$ 1.488+((2.893)* $Y_{acc}$)+((2.921)* $Y_{acc}$ ^2)+((1.410)* $Y_{acc}$ ^3), $R^2$adj.= 0.79 , P<0.05. Red dotted lines show the 95% prediction intervals. Note that the Mini Buoy design differs slightly between both case studies as the Bay of Fundy design has an additional temp/light logger attached.

Predicted median current velocities within the disused aquaculture pond at the North Sumatra study site mostly ranged between 0.09 and 0.12 m/s and were thus near the detection limit of this Mini Buoy design. Two buoys, within the channel filling and draining the abandoned aquaculture pond, B5 and B8, reached median velocities of 0.13m/s (see Fig. 4 for B8 predictions). Inundation duration ranged between 292 and 635 min/d within the abandoned pond (Fig. 5b). The spatial pattern of inundation





290 duration showed that drainage occurred via the tidal creek in the Northern part of the pond (see animation of inundation events online). The shorter inundation durations generally corresponded to the more densely recolonized areas of the abandoned pond, whereas the Southeast corner of the pond and the channel with inundation durations exceeding 460 min/d showed a lack of recolonization by mangrove trees (Fig. 5b). However, these are short-term assessments to test the Mini Buoy performance, and further investigation into inundation durations should be carried out over a spring/neap cycles as described in section 3.2.

295 All Mini Buoy acceleration data, temperature and light data and Aquadopp data are available online.

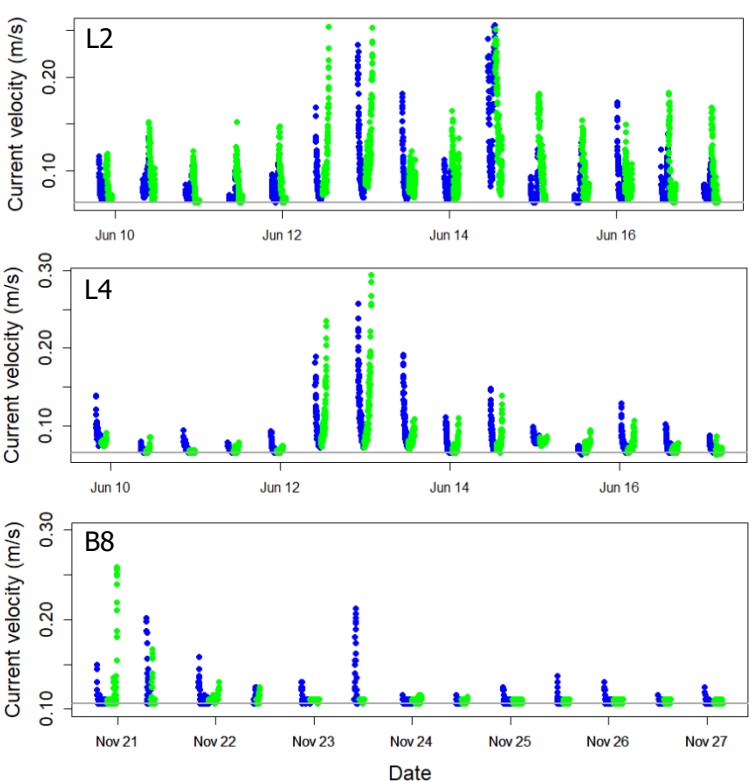

**Figure 4: Examples of predicted current velocities during predicted inundation events for L2, L4 and B8 using the LDA (Fig. 3a) for inundation and regression analysis for velocity predictions (Fig. 3b). Colours denote the first and second half of the inundation time period (approx. flood and ebb tide) and the grey line indicates the detection threshold (i.e. Bay of Fundy (L2, L3) = 0.06 m/s, North Sumatra (B8) = 0.106 m/s, see Fig. 3b).**


## A) Bay of Fundy

## B) North Sumatra

Flooding duration (min/day)

Figure 5: a) Summary of predicted median current velocities and inundation duration for the entire monitoring duration (based on
1-minute aggregated data) along the elevational gradient of the terraced salt marsh at the Bay of Fundy. b) Inundation duration at
the logger locations at North Sumatra field site.

## 3.2 Hydrological site assessments for mangrove restoration in SE Asia using the Mini Buoy application


The developed R shiny application operates with acceleration data over 15-minute intervals using 10-second interval
acceleration measurements to allow deployments of several months. The LDA using 10-second interval acceleration data for
15-minute time step predictions achieved an apparent error rate of 0.017 for Mini Buoys B11 and B12 (appendix A2). A fixed
threshold of -0.5g median y-axis acceleration to separate inundated and non-inundated events can be applied to replace the
LDA predictions. With a fixed threshold, inundation events were wrongly classified 2.49% of the time when compared with





LDA predictions for buoys B11 and B12. Correlation between y acceleration measured by the Mini Buoys, and median current velocities measured by the Aquadopp current meter in 15-minute intervals was best explained using linear regression ($V_{cur}$ = 1.173+1.059* $Y_{acc,}$ Radj = 0.7724, P<0.05). This low-frequency calibration matches the calibration against 1Hz acceleration data for values within the calibration dataset (appendix A2b). This 15-minute fixed threshold analysis as described above was

implemented in the R shiny app to allow deployment over longer durations and without calibration or training datasets and is aimed at mangrove restoration practitioners in SE Asia. We further applied the Mini Buoy and the R shiny app across 5 different mangrove settings in North Sumatra (Fig. 6). None of the deployments on tidal flats within abandoned aquaculture ponds ('Abandoned 1', 'Abandoned 2' and 'Restoration' site deployments) recorded median current velocities or 75th percentile velocities above the detection limit. Only the short-term open coast and the tidal channel deployments showed 75th

percentile velocities of up to 0.2 and 0.17 m/s, respectively. Site 'Abandoned 1' had prolonged periods of stagnant water (i.e. low frequency and long inundation periods) due to a sluice system partially in operation during the monitoring period, whereas sites 'Abandoned 2' (i.e. 'long-term' in Fig. 2) and 'Restoration' could freely flood and drain through openings in the embankment. Only the 'restoration' site showed inundation free Windows of Opportunities (WoO) for more than two consecutive tides during the monitoring period.


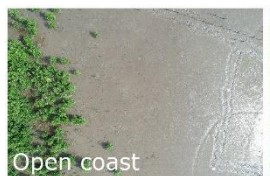
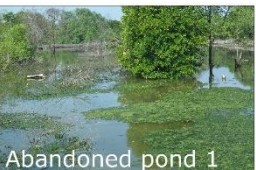
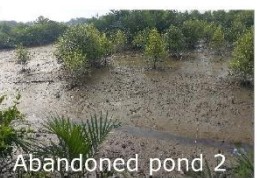
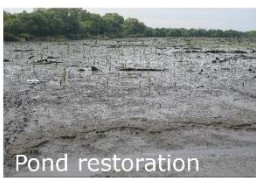
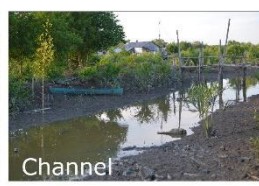

| Indicator | Open coast | Abandoned 1 | Abandoned 2 | Restoration | Channel |
|---|---|---|---|---|---|
| Monitoring period (d) | 6.25 | 50.15 | 50.64 | 47.45 | 6.64 |
| Average high tide duration (min) | 310.00 | 6492.00 | 282.34 | 218.29 | 343.85 |
| Flooding frequency (d⁻¹) | 1.92 | 0.20 | 1.90 | 1.60 | 1.96 |
| Maximum WoO duration (d) | 0.36 | 0.91 | 0.84 | 3.64 | 0.34 |
| Average flooding duration (min/d) | 624.06 | 1298.52 | 556.82 | 373.38 | 702.99 |
| Median current velocity (m/s) | 0.14 | 0.11 | 0.11 | 0.09 | 0.13 |
| 75 percentile velocity (m/s) | 0.20 | 0.11 | 0.11 | 0.09 | 0.17 |
| Flood-ebb median velocity (m/s) | 0.01 | 0.00 | 0.00 | 0.00 | -0.01 |


**Figure. 6: Summary of site characteristics in North Sumatra using the Mini Buoy and the R shiny application. Note that 'Abandoned 2' corresponds with the long-term location and 'Channel' corresponds with B1 in Fig. 2. Only 'Open coast' and 'Channel' show current velocities above the detection threshold. At site 'Abandoned 1' a sluice system was partially in operation leading to prolonged non-tidal flooding.**






## 4 Discussion

We successfully designed and calibrated the Mini Buoy, a small bottom-mounted float equipped with acceleration data logger,
in contrasting intertidal environments. We further developed the Mini Buoy into a ready-to-use tool for mangrove restoration
hydrological site assessments with a focus on disused aquaculture ponds and unvegetated tidal flats in SE Asia. This study
provides i) a design concept for low-cost monitoring of abiotic parameters in intertidal environments and ii) enables
practitioners to directly implement the Mini Buoy using an R shiny application in mangrove restoration, rehabilitation, and
creation projects in SE Asia. Calibrating the acceleration data of the Mini Buoy against water-level and current-velocity data
showed that current velocities in excess of 0.1 m/s (>0.06m/s with added temp/light logger at the Bay of Fundy) could be
detected. Inundation events could be correctly identified using the Mini Buoy, with misclassifications below 1.6% and 2.5%
when using a machine learning approach or a fixed vertical acceleration threshold respectively. With measurement intervals
of 10 seconds, the Mini Buoys as described here can be deployed for several months before data retrieval and battery recharging
becomes necessary (the authors have successfully deployed the Mini Buoy for 164 days in a separate study using the design
presented here). Whilst the Mini Buoys were able to represent a wide range of flood-ebb conditions, the current design has a
limited capacity to measure low current velocities (<0.1 m/s) and cannot make current velocity predictions when the Mini
Buoy is partially submerged (i.e. when water depths are below or around 20 cm). The latter issue is resolved in the R shiny
app by implementing an algorithm that detects individual inundation events and can identify non-tidal flooding events (e.g.
due to sluice gate operations). Moreover, a buffer period around the start and end of each inundation (1 hour in the R shiny
app) is deleted to ensure that no erroneous current velocity predictions for partially inundated loggers are made. Despite the
lower accuracy of current velocity information provided by the Mini Buoys compared to conventional hydrodynamic
monitoring equipment, the low costs and quick direct deployment and analysis (e.g. no levelling or specialist hydrological
knowledge required) make this a suitable tool to assess hydrological site conditions at ecologically relevant scales. Additional
loggers used in combination with the Mini Buoy can provide further insight into temporal variability of abiotic conditions in
combination with tidal flooding (see flood and ebb temperature plot in appendix A3). Calibrations in two very contrasting
conditions (i.e. Bay of Fundy and North Sumatra) show that the concept design is robust and easy to implement even in strong
currents, humid and hot climates and on soft muddy tidal flats. We demonstrated that the single-axis acceleration tilt calculation
principle is of sufficient accuracy to identify ecologically relevant inundation and current velocity information. Hence it is
possible to reduce the costs of the Mini Buoy even further.


The primary information that the Mini Buoys provide are in-situ flooding characteristics. This data can be compared to species-
specific flooding thresholds from the literature. Van Loon et al., (2007), for example, estimated from a 1-month flooding time





series in the Mekong delta, that flooding duration was between 800 and 400 min/d for mangrove pioneers *Avicennnia* spp. and *Sonneratia* spp., whereas the *Rhizophora* spp. / *Ceriops* spp. / *Brugueira* spp. zone had 400-100 min/d inundation durations

on average. In addition to the physiological tolerance to flooding of established plants, mangrove propagules of *Avicennia* spp. require approximately 1-5 inundation free days to sufficiently root against dislodgement by the tide on bare tidal flats (i.e. Window of Opportunity (WoO) (Balke et al., 2015). The stronger the hydrodynamic forcing upon first submergence, the longer the required WoO (i.e. the longer the roots required to stay anchored). The Mini Buoy is well suited as a tool to identify WoO as it measures inundation and hydrodynamic forcing directly on the tidal flat where new establishment may occur. Whereas

WoO may be very stochastic and require deployment across seasons or even years to detect, average inundation conditions and velocities can be adequately estimated with monitoring across one or more spring-neap tidal cycle (>15days). To allow for a first rapid assessment of hydrological site suitability without expert knowledge and with a single Mini Buoy the R shiny app contains several warning/interpretation messages related to SE Asian mangrove habitat suitability. For general site unsuitability (WoO < 1 day and for inundation durations >800 min/day or <100 min/d) warning messages will appear.

Velocities are classified as low (≤15cm/s) and high (> 15cm/s) to allow a first estimation of hydrodynamic exposure. Warning messages will also appear where measurements do not cover spring and neap tidal cycles (i.e. are shorter than 15 days). For more detailed single-deployment approaches, Mini Buoy data could be combined with a parameterized WoO model of the local pioneer species including root-growth rates and stability thresholds (Balke et al., 2014, 2015). Wave exposure could be included in site suitability assessments for wave exposed tidal flats, however, such sites are generally not recommended for

restoration in the first place due to low success rates (Wodehouse and Rayment, 2019).

Our mangrove site comparison (Fig. 6) using the Mini Buoys demonstrated how hydrological site conditions can vary locally. The site 'Abandoned 1' was the only site which exceeded the 800 min/d inundation threshold due to a temporarily closed sluice gate in the second half of the monitoring period. Hence the data from the Mini Buoy suggested that the remaining

mangrove trees in 'Abandoned 1' (see photo in Fig. 6) will die off if the present flooding regime is maintained. Flooding duration in the 'Abandoned 2' site was generally suitable for mangrove pioneer colonization and current velocities were below the detection limit. This was corroborated by the recent establishment of *Avicennia* spp. saplings and trees at the site. The 'Restoration' site assessment confirmed that previously planted *Rhizophora apiculata* at this site is likely to survive as the flooding duration is slightly below 400 min/d and no significant current velocities were detected. This was the only site where

WoO of >1 day were detected during the monitoring period, which suggests that additional natural colonization could be expected if dispersal is not hindered. The predicted current velocities of the 'Open coast' tidal flat location (Fig. 6) showed the typical tidal pattern with flood, slack and ebb tides (see Fig. A4).

Physiological tolerance to tidal inundation and establishment thresholds of mangroves also depend on other environmental factors such as soil anoxia, soil porewater salinity, bioturbation, microclimate etc. (Krauss et al., 2008). These factors may

enhance or restrict the flooding tolerance of mangrove seedlings and trees at a given location. We therefore advocate the deployment of two Mini Buoys in tandem (as implemented in the R shiny application) in order to generate site-specific





physiological thresholds, rather than rely on published information alone. This is especially recommended during short-term deployments over single spring-neap cycles. One Mini Buoy monitoring the conditions at the target restoration site can be compared to Mini Buoy data from a local reference site with the desired conditions and plant community. This has several

benefits: i) the reference site will likely experience the same regional stochastic variation in water levels (i.e. wind setup) over the monitoring period, ii) the reference site will have the same local species pool and iii) the reference site will have similar edaphic soil conditions. Reference sites should show recent seedling establishment of the desired species as conditions suitable for seedling establishment are not equal to conditions required for survival of mature mangrove trees (Krauss et al., 2008). An example of the R shiny app output for a comparative deployment between is provided in the appendix A4.

With this study we were able to show that low-cost monitoring of hydrological conditions in intertidal environments (inundation and current velocities) across multiple spring-neap tidal cycles is possible using single axis accelerometer technology and readily available materials for a bottom mounted float (i.e. the Mini Buoy). Whereas this approach can be a useful tool in coastal research, we especially highlight the Mini Buoy as a stand-alone tool for easy-to-implement hydrological site suitability assessments prior to mangrove restoration. Learning from past mistakes in mangrove restoration, where a lack

of hydrological site assessments has led to very low restoration success rates (Dale et al., 2014; Kodikara et al., 2017; Zaldivar-Jimenez et al., 2010), this new affordable and easy to implement technology will be able to assist upscaling of future mangrove rehabilitation efforts (Worthington and Spalding, 2018). The Mini Buoy can be globally implemented as a standard for mangrove restoration site assessments as it is low cost, easy to transport across borders, difficult to spot on the tidal flat, not prone to storm damage as it is close to the ground and does not require specialist knowledge in coastal engineering or data

analysis. Especially in the complex hydrological networks of abandoned aquaculture ponds in SE Asia, the Mini Buoy has the potential to efficiently assess site and pond specific conditions with only a few Mini Buoys rotated around the site. The Mini Buoy concept design and data analysis could also be applied for hydrological monitoring of river floodplain/riparian systems.






**Appendices**

Figure A1: Mini Buoy components and installation

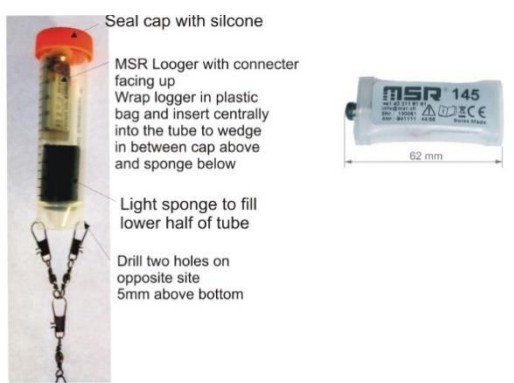

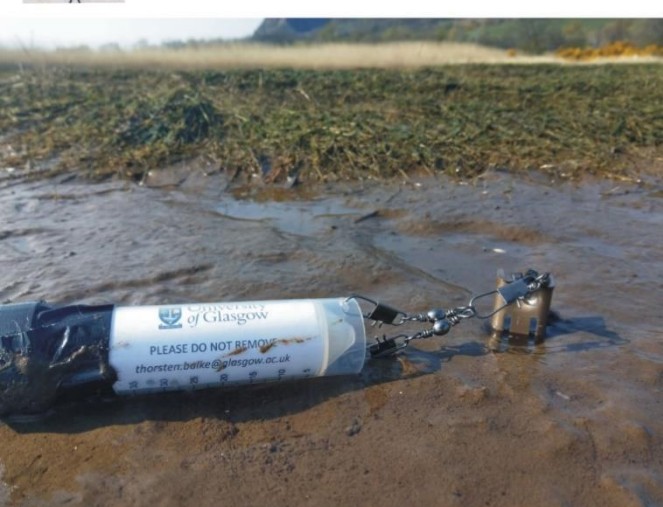

Fix logger to metal stake, protruding <1cm from tidal flat sediment


| Item | Size | Weight |
|---|---|---|
| **Corning® 50 mL PP Centrifuge Tubes, Self-Standing CentriStar™** | 114.9mm (L) | 14.6g |
| **Accelerometer MSR 145 B4A (MSR Electronics GmbH, Switzerland)** | 20 x 14 x 62 mm (W x H x L) | 18g |
| **3x Interlock Snap Fishing Swivel with Barrel (#1)** | 55mm (L) | 1.9g |
| **Foam to insert at bottom of logger, plastic bag to wrap accelerometer, tape, silicone seal** | various | < 4g |
| **TOTAL** | | 42.3g |



Figure A2 Calibration of 10 second measurement interval data for 15-minute time steps


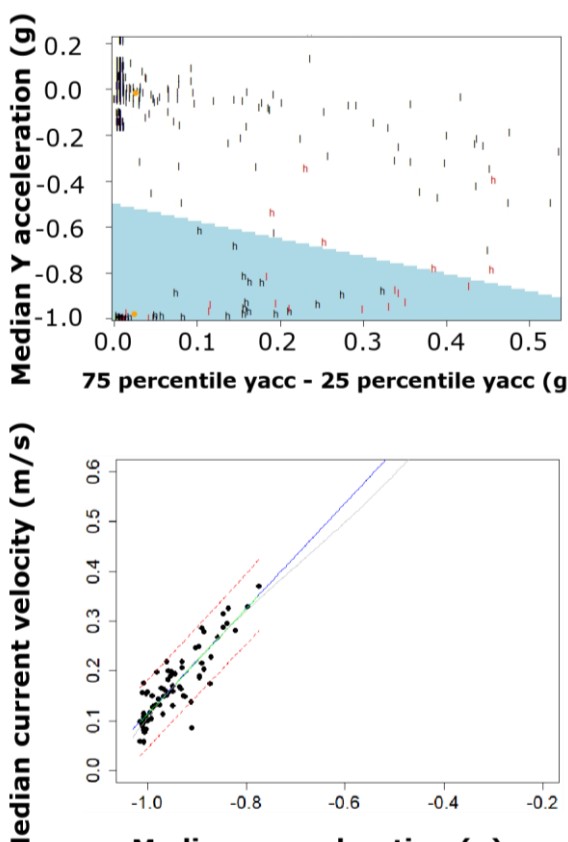

a) Linear Discriminant Analysis for North Sumatra dataset (B11 and B12) using 10-second interval acceleration data and aggregation over 15-minute time steps. The LDA group averages are indicated as orange dots. The -0.5 y acceleration threshold

to separate inundation from non-inundation time steps was applied for the R shiny application.

Linear regression (blue line) provided the best fit for the median y-axis acceleration [$Y_{acc}$ (g)] against the measured current velocity of B1 [$V_{cur}$ (m/s)] for 15-minute intervals. $V_{cur} = 1.173 + 1.059* Y_{acc}$, Radj $= 0.7724$, $P < 0.05$. The grey line shows the 1Hz deployment regression line of Fig. 3 and the green line the predictions within the measured range of values.





Figure A3 Time series example of temperature data at the Bay of Fundy Mini Buoy L2.

Measured temperature during inundation by the Hobo data logger fixed to the top of the Mini Buoy L2, colours indicate first and second half of inundation.

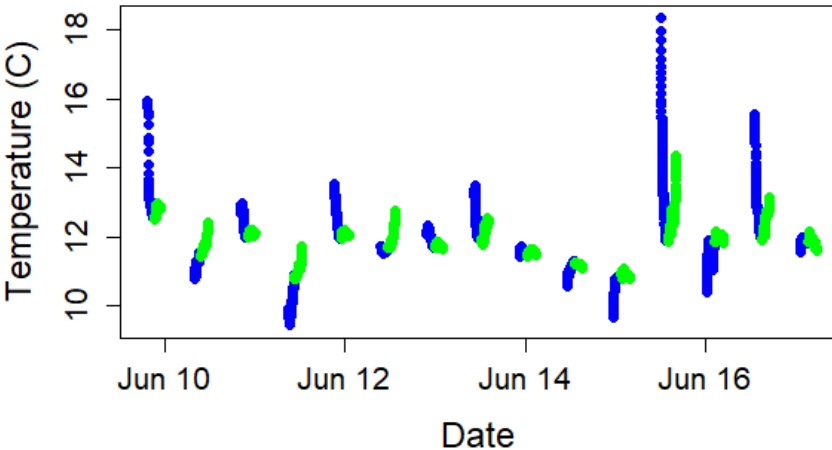

Figure A4 Comparative deployment example output of the Mini Buoy R shiny application. Site 'Restoration' at Belawan was used as the site of interest and the reference site was the 'Open coast' deployment (Fig. 6).

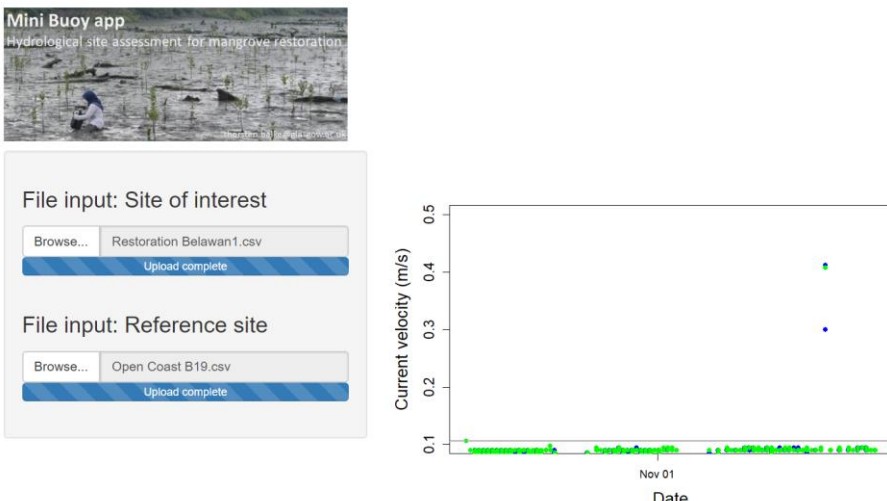






| Indicator | Value |
|---|---|
| Monitoring period (d) | 47.45 |
| Avg. high tide duration (min) | 218.29 |
| Flooding frequ. (F/d) | 1.60 |
| Max. WoO duration (d) | 3.64 |
| Avg. flooding duration (min/d) | 373.38 |
| Median current vel. (m/s) | 0.09 |
| 75 percentile velocity (m/s) | 0.09 |
| Flood-ebb median velocity (m/s) | 0.00 |

Site of interest - Interpretation: Inundation is tidal - Deployment duration OK - current velocities below detection limit - Inundation duration generally suitable for mangroves - WoO detected - Low current velocities

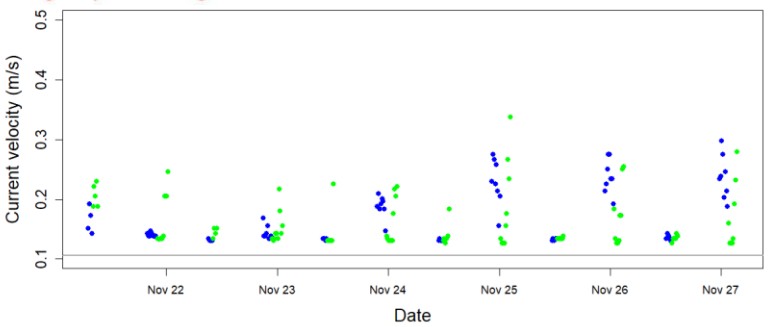

| Indicator | Value |
|---|---|
| Monitoring period (d) | 6.25 |
| Avg. high tide duration (min) | 310.00 |
| Flooding frequ. (F/d) | 1.92 |
| Max. WoO duration (d) | 0.36 |
| Avg. flooding duration (min/d) | 624.06 |
| Median current vel. (m/s) | 0.14 |
| 75 percentile velocity (m/s) | 0.20 |
| Flood-ebb median velocity (m/s) | 0.01 |

Reference site - Interpretation: Inundation is tidal - Deployment should be longer - Currents detected OK - Inundation duration generally suitable for mangroves - No WoO detected - High current velocities

Difference of conditions: Site of interest - reference site

| Indicator | Difference |
|---|---|
| Monitoring period (d) | 41.20 |
| Avg. high tide duration (min) | -91.71 |
| Flooding frequ. (F/d) | -0.32 |
| Max. WoO duration (d) | 3.27 |
| Avg. flooding duration (min/d) | -250.69 |
| Median current vel. (m/s) | -0.05 |
| 75 percentile velocity (m/s) | -0.11 |
| Flood-ebb median velocity (m/s) | -0.01 |






**Code and data availability:**

The Rshiny app can be used directly online under: https://mangroverestoration.shinyapps.io/MiniBuoyApp/

Code is available on Github: https://github.com/thorstenbalke/Mini-Buoy.git

1. Mini Buoy R shiny app code (.R)

2. Mini Buoy Manual

Zenodo repository: 10.5281/zenodo.4244642

1. Readme file

2. Aquadopp data (.dia):

2.1 North Sumatra

2.2 Bay of Fundy

3 Mini Buoy data (.csv):

3.1 Bay of Fundy: Mini Buoy acceleration L1-8 and Temperature/Light data

3.2 North Sumatra study site: Mini Buoy acceleration B1-16

3.3 North Sumatra additional sites (.csv)

- Open Coast Percut Sei Tuan

- Abandondend pond 1 Percut Sei Tuan non-tidal

- Abandonded pond 2 Percut Sei Tuan (study site)

- Restoration Belawan

- Channel (B1)

4 Animations of inundation/velocity predictions (mp4, gif):

4.1 Bay of Fundy animation of current velocities at 10-minute time steps, arbitrary scale for the bars (.gif)

4.2 Inundation animation of the North Sumatra field site for two tidal cycles: (.mp4)

**Author contribution**

Balke designed the Mini Buoy, carried out the data analysis and coding of the R shiny app and lead the writing of the manuscript. Balke and Basyuni carried out fieldwork in Indonesia. Balke and Chmura carried out fieldwork at the Bay of Fundy. Schwarz assisted with hydrodynamic analyses for the Bay of Fundy dataset. Balke, Vovides, Schwarz, Chmura, Ladd and Basyuni jointly wrote and improved the final manuscript.



## Acknowledgements

Balke and Vovides acknowledge NERC grant NE/P014127/1 for financial support. Balke acknowledges the Early Career Mobility Scheme of the University of Glasgow and the British Society for Geomorphology for additional financial support to conduct fieldwork at the Bay of Fundy. Balke and Ladd acknowledge Living Deltas NE/S008926/1. Basyuni acknowledge DIPI-LPDP grant NE/P014127.1 for financial support. We thank M. Kalacska for loan of the Emlid DGPS Rover. We thank Bejo Slamet, Nurdin Sulistiyono, Yunta Bimantra, Riska Amelia, Muttia Chandra, and Nur Indah Lestari for help with

fieldwork at Percut Sei Tuan. We thank Zulkifli and Legiman for allowing us to access the former aquaculture ponds at Percut Sei Tuan. We thank Yagasu for allowing us to access their restoration site at Belawan.

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
