# Peer review of "Monitoring tidal hydrology in coastal wetlands with the 'Mini Buoy': applications for mangrove restoration"

_Hydrology and Earth System Sciences, 2020_

## Referee Comment (RC1) · Anonymous Referee #1 · 3 Jan 2021

Review of "Monitoring tidal hydrology in coastal wetlands with the 'Mini Buoy' : application for mangrove restoration" by Thorsten Balke et al.

This study presents a novel yet low-cost bottom-mounted float Mini Buoy that can be used to monitor tidal inundation characteristics and current velocities based on the single-axis equilibrium acceleration principle. As far as I know, the facility is particularly useful for deriving the key parameters such as submersion time and current velocities, which can further be used for hydrodynamics analysis and mangrove restoration planning. Generally, this is a good study and I read it with great interests. However, the quality of the paper can be improved if the authors can properly address the following major and minor concerns.

1) I would suggest the authors to reshape the paper structure in order to improve the readability. Specifically, the different subsections in the Introduction part can be integrated to highlight the studied topic, the existing problem and the solution etc. With regard to the Method part, I would suggest the authors to separate the 'Field sites', while the rest parts were used to detail the adopted facility and how the results were generated. Finally, the Discussion part is rather long, while the Conclusions part is missing.

2) Subsections 2.3 and 2.4: it appears that the authors include some results in the method section. I would suggest the authors to move these parts to the Results section.

3) It is worth noting that the deployment period of the Mini Buoys in both field sites are less than a typical spring-neap cycle (approximately 15 days). What's the potential influence of deployment period on the calibration against the observed current velocities? In general, if one would like to use the Mini Buoys in their own studied sites, suitable calibration against observed velocities using ADCP is rather critical. The longer the measurements of current velocities, the better the calibration of the Mini Buoys, am I right?

4) The Bay of Fundy Mini Buoys were fitted with a temperature and light logger. What's the purpose? And Does these additional parameters help to set a scientific guidelines for mangrove restoration planning?

5) Appendices: some of the materials can be moved to the supplemental material. In addition, the arrangement of each figure can be improved to have a better readability. For instance, in Figure A1, the authors mixed the Figure and Table together.

Some minor comments:

1) Line 53: Here you only need define the "Windows of Opportunity (WoO)" once, for the rest you could directly refer to "WoO" (such as Lines 363 and 412).

2) Lines 280-285: the format of the equations should follow the journal's requirements. Such as the "$Y_{acc}$^3" should be replaced by "$Y_{acc}^3$", the "$R^2$adj." should be replaced by "$R^2_{adj.}$" etc.

3) Figure 3A: It is difficult to immediately understand the key points.

4) Figure 4: legends can be added.

5) Figure 6: It is better to separate the Table from the Figure. In the table, it is not

necessary to show the numerical data of "Average high tide duration (min)" and the "Average flooding duration (min/d)" with too much accuracy (i.e., integral would be enough).

---

## Referee Comment (RC2) · Anonymous Referee #2 · 9 Jan 2021

General comments

The device that is presented in this paper is novel, and very relevant for the restoration of coastal wetlands. The paper is easy to read and well structured. I have a few minor comments that could easily be addressed. But otherwise, I think this paper is suitable for publication, and hopefully the mini-buoy will be implemented in restoration projects as soon as possible.

Specific comments

Title: The initial thought that came to mind thinking of a mini-buoy, was a surface buoy. The deployment and functioning of the device is well explained. However, since the

mini-buoy is meant for deployment in restoration projects in SE Asia where buoys and other visible devices are often stolen (as already pointed out by the authors), a more intuitive name such as mini-mooring might not scare managers off purely based on the name. Alternatively, add a short description of the device or that it is submerged to the title.

L49: why specifically for SE Asian mangrove restoration? From the abstract I understand that the buoy has been deployed in an abandoned aquaculture pond system in Sumatra. But the specific application for SE Asian mangroves seems out of the blue here in the intro. I miss the link between the need for hydrology assessment and why this is specifically applicable to SE Asia in the intro. A general description of aquaculture hydrology, importance of aquaculture in terms of surface area and why abandoned aquaculture is interesting for mangrove restoration would be useful background information for a wider audience. In addition, I think that the mini-buoy could also be useful at many other target restoration sites like de-embanked polders and saltpans. Why the focus on aquaculture?

L195: "The Mini Buoys were considered 'flooded' when fully submerged (i.e. bed elevation + 20 cm). " → So sites without inundation detection in the app could in fact still be inundated, just with less than 20 cm of water, which might still be a significant amount of water for a seedling.

L219: "influence of outliers: (e.g. due to passing boats, waves or turbulence)." Passing boats would indeed create outliers. But at ocean facing sites I can imagine that waves have a large influence on the current velocities measured by the mini-buoy in a more regular manner. How was the effect of waves handled / are the current velocities by waves included in the net current velocity reported?

L 415: The authors acknowledge that long term deployment of the buoy would capture more hydrological accuracy, though the average conditions could be estimated within a spring neap cycle. I think that it would be could to mention that the timing

of that short measurement period matters, especially if ecological mangrove restoration is the target. Especially in the java sea, inundation free periods can vary greatly in length throughout the year. There can be a seasonal difference in average water level of 10 cm (see local tide stations for long term fluctuations http://www.ioc-sealevelmonitoring.org/list.php ), driven by the monsoon winds. It is important to address that the hydrodynamic characteristics of an intended restoration site should especially be sufficient during the fruiting season of the targeted species.

Figure 5: What is the explanation for the velocity minimum at L3, and subsequent increase in current velocities at higher elevations? Could that be an effect of lower accuracy when the water levels became lower, or is it an expected effect at this site. Why is there no velocity graph for figure 5b?

L462:" The Mini Buoy concept design and data analysis could also be applied for hydrological monitoring of river floodplain/riparian systems." seems a rather offhand comment, not a nice wrap up of the story or take home message.

Technical corrections

Figure 4: blue being flood and green being ebb? Figure 5: b, scale numbers and units are hard to read. Figure 6: very nice to see an example of an aquaculture pond with stagnant water and partially operational sluice system in here as an example of a very unsuitable site for restoration Figure A2: Orange letters instead of orange dots?

---

## Author Comment (AC1) · 8 Feb 2021

Reviewer #1 comment:

This study presents a novel yet low-cost bottom-mounted float Mini Buoy that can be used to monitor tidal inundation characteristics and current velocities based on the single-axis equilibrium acceleration principle. As far as I know, the facility is particularly useful for deriving the key parameters such as submersion time and current velocities, which can further be used for hydrodynamics analysis and mangrove restoration planning. Generally, this is a good study and I read it with great interests.

Author reply: We would like to thank the reviewer for their time and valuable comments. We have addressed them individually below.

1) I would suggest the authors to reshape the paper structure in order to improve the readability. Specifically, the different subsections in the Introduction part can be integrated to highlight the studied topic, the existing problem and the solution etc.

Author's reply: In this manuscript we have tried to achieve a balance between introducing a new method and discussing two case studies which may have led to a less integrated structure in the introduction. We have addressed this by restructuring the introduction and adding a new section 1.2 more specifically on mangrove restoration and aquaculture ponds LL 46-58.

'1.2 Hydrological and hydrodynamic bottlenecks to mangrove restoration

Assessing the the local hydrology prior to mangrove restoration is needed to determine whether conditions are too harsh for seedlings to survive and need to be mitigated (Albers and Schmitt, 2015) or whether insufficient flooding may lead to hypersalinity or succession towards terrestrial plant communities (Lewis, 2005). One of the main reasons for mangrove deforestation in the past, and hence one of the major opportunities for mangrove restoration today, are aquaculture ponds (Dale et al., 2014; Primavera and Esteban, 2008). Breaching the embankments of such ponds, built within previous mangrove forests, is an effective way to restore mangrove forests. However, previous excavation of sediment and altered tidal channels and remaining weirs may have altered the hydrology. Hence spatially explicit monitoring of inundation durations are required prior to attempting mangrove restoration (Dale et al., 2014). Especially stagnant water and inhibited drainage can locally create restoration bottlenecks in abandoned tidal pond complexes. At hydrodynamically exposed sites, where current velocities are able to dislodge recently rooted mangrove propagules, inundation free periods of several days provide a Window of Opportunity (WoO) to surpass critical establishment thresholds (Balke et al., 2011, 2014). WoO are particularly important to assess in unassisted restoration projects to predict suitability for natural colonization by mangrove pioneer species.'

With regard to the Method part, I would suggest the authors to separate the 'Field sites', while the rest parts were used to detail the adopted facility and how the results were generated.

Author reply: We have now explicitly separated the Indonesian field sites into 'Short-term deployment sites' and 'Long-term deployment sites'.

Finally, the Discussion part is rather long, while the Conclusions part is missing.

Author reply: We have moved the final part of the discussion into a separate conclusions paragraph and further changed the text to provide more relevant closing statement as suggested by reviewer #2 LL 640-651.

'4 Conclusions

With this study we were able to show that low-cost monitoring of hydrological conditions in intertidal environments (inundation and current velocities) across multiple spring-neap tidal cycles is possible using single axis accelerometer technology and readily available materials for an underwater float (i.e. the Mini Buoy). Whereas this approach can be a useful tool in coastal research, we especially highlight the Mini Buoy as a stand-alone tool for easy-to-implement hydrological site suitability assessments prior to mangrove restoration. The Mini Buoy can be globally implemented as a standard for mangrove restoration site assessments as it is low cost, easy to transport across borders, difficult to spot on the tidal flat, not prone to storm damage as it is close to the ground and does not require specialist knowledge in coastal engineering or data analysis. Especially in the complex hydrological networks of abandoned aquaculture ponds in SE Asia, the Mini Buoy has the potential to efficiently assess site and pond specific conditions with only a few Mini Buoys rotated around the site. Learning from past mistakes in mangrove restoration, where a lack of hydrological site assessments has led to very low restoration success rates (Dale et al., 2014; Kodikara et al., 2017; Zaldivar-Jimenez et al., 2010), this new affordable and easy to implement technology will be able to assist urgently needed upscaling of future mangrove rehabilitation efforts (Worthington and Spalding, 2018).'

2) Subsections 2.3 and 2.4: it appears that the authors include some results in the method section. I would suggest the authors to move these parts to the Results section.

Author reply: We agree and have moved the description of the Rshiny application into the results section under 3.2.1. L463ff.

3) It is worth noting that the deployment period of the Mini Buoys in both field sites are less than a typical spring-neap cycle (approximately 15 days). What's the potential influence of deployment period on the calibration against the observed current velocities? In general, if one would like to use the Mini Buoys in their own studied sites, suitable calibration against observed velocities using ADCP is rather critical. The longer the measurements of current velocities, the better the calibration of the Mini Buoys, am I right?

Author reply: Successful calibration of the Mini Buoy against current velocities would benefit from 'as long as possible calibration' but does not necessarily require a full spring-neap cycle as current velocities vary between slack and flood/ebb tide even within a single tidal cycle to create a calibration curve against the dip of the Mini Buoy. We would recommend, however, to create a calibration data set that covers parts of the spring tide though as this will produce the highest ebb and flood currents in addition to the calibration nearer the slack tide. In our case we the maximum storage capacity of the current meter determined the maximum calibration duration.

4) The Bay of Fundy Mini Buoys were fitted with a temperature and light logger. What's the purpose? And Does these additional parameters help to set a scientific guidelines for mangrove restoration planning?

Author reply: Light availability under water may be important for mangrove seedlings that are flooded for long durations. This will depend on water depth and turbidity, but the effect is not well studied so far, for example compared to light requirements for seagrass. We

tested this application mainly to highlight the potential for further research purposes in relation to water quality, sediment concentrations and light availability. Added L580' for example in water quality studies'

5) Appendices: some of the materials can be moved to the supplemental material. In addition, the arrangement of each figure can be improved to have a better readability. For instance, in Figure A1, the authors mixed the Figure and Table together.
Author reply: We agree and have extracted the Table from Figure A1 to Table A2 and moved the Mini Buoy app example screenshots (previously A4) to the supplementary material deposited at zenodo.

Some minor comments:
1) Line 53: Here you only need define the "Windows of Opportunity (WoO)" once, for the rest you could directly refer to "WoO" (such as Lines 363 and 412).
Author reply: We have removed this now after first mention.

2) Lines 280-285: the format of the equations should follow the journal's requirements. Such as the "Yacc^3" should be replaced by "Yacc3", the "R2adj." should be replaced by "R2adj." etc.
Author reply: Thank you, done.

3) Figure 3A: It is difficult to immediately understand the key points.
Author reply: We have changed parts of the caption to clarify: 'Linear Discriminant Analysis (LDA) with prediction of flooded and non-flooded time steps of L2-L5 (Bay of Fundy, left) and B9, B11, B12, B14 (North Sumatra, right) based on acceleration data and validated against measured water levels.'

4) Figure 4: legends can be added.
Author's reply: We have added the legend for flood and ebb tide.

5) Figure 6: It is better to separate the Table from the Figure. In the table, it is not necessary to show the numerical data of "Average high tide duration (min)" and the "Average flooding duration (min/d)" with too much accuracy (i.e., integral would be enough).
Author reply: Agreed and done.

---

## Author Comment (AC2) · 8 Feb 2021

Reviewer #2 comments:

General comments
The device that is presented in this paper is novel, and very relevant for the restoration of coastal wetlands. The paper is easy to read and well structured. I have a few minor comments that could easily be addressed. But otherwise, I think this paper is suitable for publication, and hopefully the mini-buoy will be implemented in restoration projects as soon as possible.

Author reply: We would like to thank the reviewer for their time and valuable comments. We have addressed them individually below.

Specific comments
Title: The initial thought that came to mind thinking of a mini-buoy, was a surface buoy. The deployment and functioning of the device is well explained. However, since the mini-buoy is meant for deployment in restoration projects in SE Asia where buoys and other visible devices are often stolen (as already pointed out by the authors), a more intuitive name such as mini-mooring might not scare managers off purely based on the name. Alternatively, add a short description of the device or that it is submerged to the title.
Author reply: Thanks, we agree that the term 'buoy' most commonly refers to surface buoys although this can sometimes also refer to underwater buoys. We have made it clearer at the start that this is indeed an underwater buoy and not a surface float but decided to keep the overall name.
 L17: changed to 'underwater float'

L49: why specifically for SE Asian mangrove restoration? From the abstract I understand that the buoy has been deployed in an abandoned aquaculture pond system in Sumatra. But the specific application for SE Asian mangroves seems out of the blue here in the intro. I miss the link between the need for hydrology assessment and why this is specifically applicable to SE Asia in the intro. A general description of aquaculture hydrology, importance of aquaculture in terms of surface area and why abandoned aquaculture is interesting for mangrove restoration would be useful background information for a wider audience. In addition, I think that the mini-buoy could also be useful at many other target restoration sites like de-embanked polders and saltpans. Why the focus on aquaculture?
Author reply: Whereas the Mini Buoy can be applied on any tidal flat, we have customized the R shiny application to be used in SE Asian mangrove restoration which often includes restoration of disused aquaculture ponds.  This context has been made more explicit in the introduction:
LL 46-58:

'1.2 Hydrological and hydrodynamic bottlenecks to mangrove restoration
Assessing the the local hydrology prior to mangrove restoration is needed to determine whether conditions are too harsh for seedlings to survive and need to be mitigated (Albers and Schmitt, 2015) or whether insufficient flooding may lead to hypersalinity or succession towards terrestrial plant communities (Lewis, 2005). One of the main reasons for mangrove deforestation in the past, and hence one of the major opportunities for mangrove restoration today, are aquaculture ponds (Dale et al., 2014; Primavera and Esteban, 2008). Breaching the embankments of such ponds, built within previous mangrove forests, is an effective way to restore mangrove forests. However, previous excavation of sediment and altered tidal channels and remaining weirs may have altered the hydrology. Hence spatially explicit monitoring of inundation durations are required prior to attempting mangrove restoration (Dale et al., 2014). Especially stagnant water and inhibited drainage can locally create

restoration bottlenecks in abandoned tidal pond complexes. At hydrodynamically exposed sites, where current velocities are able to dislodge recently rooted mangrove propagules, inundation free periods of several days provide a Window of Opportunity (WoO) to surpass critical establishment thresholds (Balke et al., 2011, 2014). WoO are particularly important to assess in unassisted restoration projects to predict suitability for natural colonization by mangrove pioneer species.'

L159: 'for ecological mangrove restoration planning with reference to habitat requirements of SE Asian mangrove species.'

There are two main customizations of the app for this geographic context: i)The recorded flooding is compared against known flooding tolerance of SE Asian species in the output table for immediate interpretation by non-experts. ii) Disused aquaculture ponds often suffer from stagnant water and include a complex network of channels which maintain very shallow runoff during low tide but in which it is desirable to monitor currents during flood and ebb. We have thus decided to only include flooding of >20cm depth by using logger tilt as start and end point of flooding, neglecting standing water on the surface of the sediment. For other applications, for example to study dispersal of propagules on very shallow tidal flats this could be customized to include shallow inundation events. The code is open access so can be easily customized for further applications.
We have further clarified this in

LL: 460-485 'An online app, using the R package shiny (Chang et al., 2020), was created as a quick and easy way to assess local hydrological site conditions prior to restoration of mangroves with reference to requirements of SE Asian mangrove species. Acceleration data acquisition can be reduced to 10-second intervals to allow for longer deployment periods across spring-neap tidal cycles and seasons. For calibration of low the frequency data, Mini Buoys B1, B11 and B12 were re-sampled at 10 second intervals. The same analyses for inundation predictions using the LDA and correlation against current velocities (as described above) were carried out over 15-minute time periods. Mini Buoys were considered submerged in the calibration time series when measured water levels were above the Mini Buoy for the entire 15-minute period. Instead of the LDA, a fixed acceleration threshold was applied to differentiate between flooded and non-flooded time steps. This threshold allows for the use of the Mini Buoy and the R shiny application without the need to load training datasets and was informed by the results of the LDA. The fixed threshold predictions were further cross validated against the LDA predictions for Mini Buoys B11 and B12.
The LDA using 10-second interval acceleration data for 15-minute time step predictions achieved an apparent error rate of 0.017 for Mini Buoys B11 and B12 (Fig. A3). A fixed threshold of -0.5g median y-axis acceleration to separate inundated and non-inundated events was applied to replace the LDA predictions. With a fixed threshold, inundation events were wrongly classified 2.49% of the time when compared with LDA predictions for buoys B11 and B12. Correlation between y acceleration measured by the Mini Buoys, and median current velocities measured by the Aquadopp current meter in 15-minute intervals was best explained using linear regression (Vcur = 1.173+1.059* Yacc, $R_{adj}$ = 0.7724, P<0.05). This low-frequency calibration matches the calibration against 1Hz acceleration data for values within the calibration dataset (Fig. A3).
The R shiny application calculates inundation and current velocity statistics for the entire monitoring period: average high tide duration (min), average flooding duration (min/d), flooding frequency (d-1), maximum WoO duration (d) = longest inundation free period, median current velocity (m/s), 75 percentile current velocity (m/s), and difference of flood - ebb median velocity (m/s)….'

L565: '. This application can be easily customized to other regions with different species requirements and flooding characteristics.'

LL597-598: 'The Mini Buoy can be further customized to include detection of very shallow inundation events on tidal flats, for example to study propagule dispersal.'

L195: "The Mini Buoys were considered 'flooded' when fully submerged (i.e. bed elevation + 20 cm). " ! So sites without inundation detection in the app could in fact
still be inundated, just with less than 20 cm of water, which might still be a significant amount of water for a seedling.

Author reply: As mentioned above, we have made this choice because of the possibility of stagnant water or very shallow runoff in channels leading to and from the disused ponds. We hope to build future applications for the Mini Buoy in an open source library where users can switch between different applications/customizations, such as the suggested shallow inundation detection.
LL597-598: 'The Mini Buoy can be further customized to include detection of very shallow inundation events on tidal flats, for example to study propagule dispersal.'

L219: "influence of outliers: (e.g. due to passing boats, waves or turbulence)." Passing boats would indeed create outliers. But at ocean facing sites I can imagine that waves have a large influence on the current velocities measured by the mini-buoy in a more regular manner. How was the effect of waves handled / are the current velocities by waves included in the net current velocity reported?

Author reply: Wave-induced orbital velocities are not included in the analysis, the current velocities measured by the Aquadopp are time-averaged and are averaged across a 75cm distance for each beam. This means smaller wave-induced velocities will not be detected. The calibration in Sumatra was carried out in a tidal channel away from any wave action.

The calibration at the Bay of Fundy did, however, include larger waves of up to 0.73m sign. wave height. Wind-waves in addition to tidal currents will create a swaying motion of the Mini Boy on top of the current velocity induced dip angle. Hence, waves will mainly influence the variability of the y-acceleration signal rather than the time averaged dip angle. We have calculated wave-induced orbital velocities from the pressure sensor data of the aquadopp and established that the standard deviation of the y-acceleration signal would need to exceed 0.1 before wave orbital velocities could be detected in the variability of the signal.

Plotting the calculated orbital velocities against the time-averaged y-acceleration for the Bay of Fundy calibration showed, that there is no systematic influence of wave orbital velocity near the bed on the average dip angle (i.e. y-axis acceleration) of the Mini Buoy on the tidal flat (see figure to the right).

[Figure]

L 415: The authors acknowledge that long term deployment of the buoy would capture more hydrological accuracy, though the average conditions could be estimated

within a spring neap cycle. I think that it would be could to mention that the timing of that short measurement period matters, especially if ecological mangrove restoration is the target. Especially in the java sea, inundation free periods can vary greatly in length throughout the year. There can be a seasonal difference in average water level of 10 cm (see local tide stations for long term fluctuations http://www.iocsealevelmonitoring.
org/list.php ), driven by the monsoon winds. It is important to address that the hydrodynamic characteristics of an intended restoration site should especially be sufficient during the fruiting season of the targeted species.

Author reply: We agree very much with this comment and the timing aspect of biology and physical conditions are one of the main motivators for the design of this tool. We have included a short sentence on the aspect of fruiting season timing in relation to the monitoring period.

L 601: 'Seasonal variability of conditions, especially with respect to the fruiting season of the desired species, should be accounted for where possible.'

Figure 5: What is the explanation for the velocity minimum at L3, and subsequent increase in current velocities at higher elevations? Could that be an effect of lower accuracy when the water levels became lower, or is it an expected effect at this site. Why is there no velocity graph for figure 5b?

Author reply: Tidal currents on the mudflat generally decrease with distance from the main channel/estuary. The subsequent velocity increase at the edge of the marsh plateau may be due to ebb and flood peaks created by the topography of the marsh edge when the marsh is about to become inundated/drained.  As mentioned in the main text, current velocities within the former aquaculture pond were mainly near the detection limit. A figure on the current velocities would therefore not provide further useful information.

L462:" The Mini Buoy concept design and data analysis could also be applied for hydrological monitoring of river floodplain/riparian systems." seems a rather offhand comment, not a nice wrap up of the story or take home message.
Author reply: We have deleted this and added a conclusion paragraph in line with comments made by Reviewer #1.

Technical corrections

Figure 4: blue being flood and green being ebb?
Author reply: We have added a legend.

Figure 5: b, scale numbers and units are hard to read.
Author reply: changed accordingly

Figure 6: very nice to see an example of an aquaculture pond with stagnant water and partially operational sluice system in here as an example of a very unsuitable site for restoration

Figure A2: Orange letters instead of orange dots?

Author reply: The orange dots were added in addition the letters to show the group averages.
Clarified in the Figure legends now.